# Effect of Various Acid Solutions as an Aid in Removing the OrthoMTA-Based Root Canal Filling

**DOI:** 10.3390/ma16134535

**Published:** 2023-06-23

**Authors:** Naveen Chhabra, Abhishek Parolia

**Affiliations:** Restorative Dentistry Division, School of Dentistry, International Medical University, Bukit Jalil, Kuala Lumpur 57000, Malaysia; abhishek_parolia@imu.edu.my

**Keywords:** endodontic retreatment, OrthoMTA, citric acid, glycolic acid

## Abstract

The objectives of this study were to compare the effects of various acid solutions combined with ultrasonics as an aid to remove mineral trioxide aggregate (MTA)-based root canal filling and to assess their effect on the surface topography and microhardness of root canal dentin. Materials and Method: Fifty human permanent single rooted and single canaled freshly extracted teeth were decoronated and sectioned apically to prepare the middle third of root sections of 5 mm length. The canals were prepared in a step-back manner. OrthoMTA was packed throughout the prepared canals. These root sections were incubated for one week and subsequently randomly allocated to five groups (n = 10) according to the OrthoMTA removal method: No treatment (NT); 5% glycolic acid + ultrasonics (5% GA+U); 10% glycolic acid + ultrasonics (10% GA+U); 10% citric acid + ultrasonics (10% CA+U); Distilled water + ultrasonics (DW+U). A 1 mm deep well was created within the coronal end of the set OrthoMTA. Wells were filled with each respective test solution and left for 5 min. Thereafter, further removal of OrthoMTA used a specific ultrasonic tip. Finally, the canals were flushed using 1 mL of the respective test solutions and activated with a Controlled Memory ultrasonic tip for two cycles of 20 s each followed by flushing with 1 mL of distilled water and paper point drying of the canals. Then, specimens were longitudinally split into two halves and examined under a scanning electron microscope (1000×) to assess the residual OrthoMTA and surface topography of root canal dentin. The Vickers surface microhardness of treated radicular dentin was measured using the HMV-2 microhardness tester. Result: Data were analysed using one-way ANOVA followed by Tukey’s post hoc test. Significant differences for residual OrthoMTA were observed between (10% GA+U) with (5% GA+U), (10% CA+U), (DW+U) and (NT) (*p* value < 0.01). In the context of microhardness, (5% GA+U) and (10% GA+U) showed statistically significant difference compared to (NT), (10% CA+U) and (DW+U) (*p* value < 0.01). Conclusion: 10% GA+U was superior to other tested groups in removing OrthoMTA, but it substantially reduced dentin microhardness.

## 1. Introduction

Mineral trioxide aggregate (MTA) has been widely used in endodontics since its introduction as a root-end filling material for apical surgery [1]. MTA is a hydrophilic powder that hydrates to form hydrated calcium silica and calcium hydroxide [2]. MTA is now successfully used in endodontics as direct pulp capping, pulpotomy, apexification, repair of root perforation and regenerative endodontic procedures, owing to its sealing ability, antibacterial activity, biocompatibility, osteo-inductive potential and its ability to bond to the tooth structure, thus providing a monobloc effect [3,4,5,6]. Its use has further expanded as an obturation material in certain demanding situations and as an intra-orifice barrier above the gutta percha to improve the coronal seal [7,8,9,10].

Recently, a newly developed OrthoMTA claims to be as biocompatible as MTA, with lower content of heavy metals. When used as a root canal filling material, it forms an interfacing layer of hydroxyapatite (Hap) between the OrthoMTA and the canal wall. It thereby prevents microleakage and induces regeneration of the apical periodontium. [7,11,12]. In addition, due to its bioactive characteristic and release of calcium ions, OrthoMTA helps with reducing periapical inflammation [13].

In the event of endodontic treatment failure, the complete removal of old obturating material is necessary. The residual filling materials may prevent contact with irrigation solutions as well as medicaments, with the persistent microorganisms hiding beneath, thus negatively affecting the long-term prognosis of retreatment [14].

Despite having many advantages, there exists a hesitancy to use OrthoMTA as a root canal filling material due to the challenges faced by clinicians in its removal during retreatment in the event of treatment failure. Numerous methods, including the use of rotary files or ultrasonic endodontic tips, have been evaluated in the past for the removal of MTA or MTA-based sealers [15]. Additionally, certain acidic solutions including 17% EDTA, 10% citric acid, 37% hydrochloric acid and 5% glycolic acid have also been tested to dissolve various MTA preparations [16,17,18,19,20]. Despite all the efforts to identify appropriate methods for MTA removal, there is no established method that is conclusive and found clinically effective. In addition, the use of strong acidic solutions and mechanical devices to remove MTA-based material or OrthoMTA may negatively influence the microhardness and surface topography of the radicular dentin.

Hence, the present study aimed to compare the effects of various acidic solutions combined with ultrasonics as an aid to remove the OrthoMTA from the root sections and assess their effect on the surface topography and microhardness of the radicular dentin. The null hypothesis proposed that test solutions will not aid in the removal of OrthoMTA, would cause significant damage to the radicular dentin and would reduce the microhardness.

## 2. Materials and Methods

This study was initiated after obtaining ethical approval from the university’s Ethics and Research Committee. Fifty human permanent single rooted and single canaled teeth extracted due to carious, periodontic or orthodontic reasons were procured from the Institution’s oral health centre according to the Ministry of Health Guidelines for Ethical Review of Clinical Research or Research Involving Human Subjects (2006). The sample size was calculated based on mean and standard deviation using G*Power 3.1.9.2. (Heinrich Heine University, Dusseldorf, Germany; Erdfelder, Faul, & Buchner, 1996), giving the power of study as 95.34% with a total sample size of 50 (10 per group). The level of significance was set at 0.05. The procured extracted teeth were cleaned, sterilized and stored in distilled water until used. All teeth were decoronated and sectioned apically using a diamond disc (NTI Flex, Kerr Dental, Brea, CA, USA) under continuous water spray to prepare the middle third of root sections of 5 mm length. The canals of the root sections were prepared with Gates-Glidden burs (Dentsply Maillefer, Ballaigues, Switzerland) from size 2 (0.70 mm) to size 6 (1.50 mm) in a step-back manner and irrigated with distilled water. The prepared root canals were irrigated with 5 mL of 3.0% NaOCl (Parcan N, Septodont, Saint-Maur-des-Fossés, France) and 17% EDTA (Largal Ultra, Septodont, Kuala Lampur, Malaysia) to remove the smear layer, followed by a final rinsing with distilled water. Apical ends of the root sections were sealed using cold cure acrylic resin (Vertex Self Curing, Sdn Bhd, Kuala Lampur, Malaysia), mimicking as an apical barrier to prevent extrusion of the OrthoMTA. OrthoMTA powder (BioMTA, Seoul, Republic of Korea) was mixed with distilled water (powder to liquid ratio of 3:1) into a paste consistency according to the manufacturer’s instructions and packed into the root sections using an OrthoMTA gun and OrthoMTA plugger (BioMTA, Seoul, Republic of Korea) with a 1.0 mm sized tip and butt end of a size F3 Protaper Gold absorbent point (Dentsply-Sirona, Charlotte NC, USA) using minimal pressure. The filled root sections were covered with a damped gauze soaked in phosphate-buffered saline and stored in an incubator with 100% humidity at 37 °C for one week to allow the complete setting of the OrthoMTA.

All specimens were randomly allocated to five groups (n = 10) according to the test solutions used for the removal of OrthoMTA. The groups were as follows—(NT): No treatment; (5% GA+U): 5% glycolic acid + ultrasonics; (10% GA+U): 10% glycolic acid + ultrasonics; (10% CA+U): 10% citric acid + ultrasonics; (DW+U): distilled water + ultrasonics. A 1 mm deep well was created using a number one round tungsten carbide bur, size 008 (Komet Dental, Lemgo, Germany) within the coronal end of the OrthoMTA for each specimen to secure the acid solution. For each of the five groups, the acid solution filled the prepared wells for all specimens and was allowed to remain there for 5 min. Thereafter, a 0.5 mm rigid ball ultrasonic tip (MTA removal Kit, Bio MTA, South Korea) was used to further remove the OrthoMTA. After the removal of the OrthoMTA, the canals were flushed using 1 mL of respective acid solutions and the 0.25mm Controlled Memory tip (MTA removal Kit, Bio MTA, Seoul, Republic of Korea) was activated for two cycles of 20 s each. Finally, all the canals were flushed with 1 mL of distilled water followed by drying using an absorbent paper point. Thereafter, each specimen was longitudinally split into two halves and examined under a scanning electron microscope (1000×) (TM3000 Tabletop Microscope, Hitachi High-Tech IPC, Kuala Lumpur, Malaysia) to assess the residual OrthoMTA and surface topography of the root canal dentin. The residual OrthoMTA was scored and tabulated according to the following criteria [21]: Score 0: 0–25% of residual OrthoMTA covering the dentinal surface; Score 1: <50% of the dentin surface covered with OrthoMTA; Score 2: 50–75% of the dentin surface covered with OrthoMTA; Score 3: 75–100% of the dentin surface covered with OrthoMTA. The SEM images were assessed by an independent evaluator and the results were tabulated according to the scoring criteria mentioned earlier.

The Vickers surface microhardness of radicular dentin of all the tested specimens was measured using the HMV-2 microhardness tester (Shimadzu, Kyoto, Japan) having a square-based, pyramid-shaped diamond indenter that produced a quadrangular depression with two equal orthogonal diagonals in the polished surface of the object. The angle between the opposite faces of the diamond indenter was set to 136 degrees. A microhardness test of dentin was performed at 0.5 mm, level from the canal lumen with a full load of 980.7 MN (HV 0.1) for 5 s at room temperature. The Vickers microhardness value was displayed on the digital readout of the microhardness tester. Two indentations were made on the dentin surface of each specimen, placed 1 mm or more apart from each other (Figure 1). For better understanding, a schematic diagram of the entire methodology is shown in Figure 2.

## 3. Results

Direct visualization of SEM micrographs suggested that (10% GA+U) resulted in the least residual OrthoMTA, followed by (5% GA+U), (10% CA+U), (DW+U) and (NT), respectively.

In group (10% GA+U), the cleanest dentinal surface was appreciable due to the presence of opened dentinal tubules and sparsely existing residual OrthoMTA particles covering 25% or less of the treated surface. Similar appearances were seen in the micrographs from (5% GA+U) and (10% CA+U); however, these groups yielded the presence of comparatively more amounts of residual OrthoMTA. The SEM images from (DW+U) showed a dentin surface entirely covered with OrthoMTA and partially eroded, accompanied with crack lines, whereas the (NT) group obviously demonstrated the presence of densely compacted OrthoMTA (Figure 3A–E).

The mean residual OrthoMTA SEM score were analyzed using ANOVA. All pair-wise treatment comparisons were performed with a Tukey adjustment for multiplicity using IBM SPSS version 26 software package (IBM Corp., Armonk, NY, USA). The significance level was set at *p* < 0.01.

Groupwise comparison displayed a statistically significant difference between (10% GA+U) with (5% GA+U), (10% CA+U), (DW+U) and (NT) (*p* value < 0.01). However, there was no significant difference observed between (5% GA+U), (10% CA+U) and (DW+U) (*p* value > 0.01) (Figure 4).

The mean value of the three measurements was calculated to determine the microhardness value for each root section. Differences between the microhardness values of dentin exposed to different test solutions and the control specimens were analyzed with a one-way ANOVA followed by Tukey’s post hoc test with IBM SPSS version 26 software package (IBM Corp., Armonk, NY, USA), with a level of significance set at *p* = 0.01.

Among these groups, (5% GA+U) and (10% GA+U) showed statistically significant difference in the dentin microhardness values in comparison to NT, (10% CA+U) and (DW+U) (*p* value < 0.01). However, the difference between group (NT), (10% CA+U) and (DW+U) and between (5% GA+U) and (10% GA+U) was statistically insignificant (*p* value > 0.01).

The mean difference values of all groups for SEM and surface microhardness tests are presented in Table 1.

## 4. Discussion

The least amount of residual OrthoMTA was seen in (10% GA+U) followed by (5% GA+U), (10% CA+U) and (DW+U), respectively, highlighting the advantage of tested solutions used as an adjunct. Boutsioukis C. et al. assessed the retreatment possibility of MTA-based obturation; however, the tested methods were not able to completely remove the MTA [15]. More recently, Soram Oh et al. evaluated the effect of various acidic solutions on the OrthoMTA and concluded that a five minute application of 10% CA and 5% GA significantly reduced the microhardness of set OrthoMTA and hold lower cellular cytotoxicity compared to 17% EDTA; however, exposure to these acids for five minutes may not be sufficient to weaken the OrthoMTA enough to facilitate its retrieval from the root canal space [20]. Therefore, elongation of exposure time to ten or twenty minutes would be beneficial to providing clinical relevance.

Boutsioukis C. et al. [15], Nandini S. et al. [16] and Butt N. et al. [17] have observed the effect of various acids including 2% carbonic acid, 20% tartaric acid, 37% hydrochloric acid, 17% EDTA and 10% citric acid on MTA and recommended the use of these acids for less than ten minutes inside the canals to prevent their deleterious effect on the mechanical properties of teeth.

Earlier, Kayahan et al. examined the matrix loss of the acid-etched surface of ProRoot MTA from 37% phosphoric acid [22], and Soram Oh et al. assessed the effect of glycolic acid and citric acid on the microhardness of OrthoMTA and its dissolution pattern. Glycolic acid and citric acid, with a pH of 2.11 and 1.59, respectively, dissolve calcium carbonate and thereby destroy the cubic crystals of acid-etched OrthoMTA [20].

The prolonged exposure to acids may be detrimental to the radicular dentin, and use of an additional mechanical method with a short duration of exposure to acidic solutions appears to be less detrimental to the dentin structure in the removal process. Hence, this study aimed to assess the synergistic effect of a chemical and mechanical method and to identify the most suitable OrthoMTA removal method where an acid solution weakened the set OrthoMTA by making it porous while the concurrent use of ultrasonic energy made it easy to dislodge from the dentinal walls [20].

There has been a long debate on the use of ultrasonics and microcrack propagation, but a study by Barakat R.M. et al. has found no significant increase in microcrack propagation after the use of ultrasonics [23]. Madarati A.A. et al. have also recommended the use of passive ultrasonics and reciprocating instruments in the removal of MTA-based obturating materials in the event of endodontic retreatment [24].

In the present study, all tested acidic solutions aided by ultrasonics removed OrthoMTA from the root sections to a greater extent, however, (10% GA+U) was found to be most effective. Glycolic acid (GA), also known as hydroxyethanoic acid or hydroxyacetic acid, is a colourless, odourless and water-soluble substance that has been found to be effective in enamel and dentin etching. However, there is a lack of consensus about the use of GA solutions and their different concentrations [5,6]. It is recommended to use caution with topical applications of 5–10% GA in the beauty industry due to its cytotoxicity in higher concentrations [25]. Soram Oh et al. found GA to be less cytotoxic in lower concentrations when compared to EDTA and CA [20]. Hence, higher concentration of GA should be used cautiously, which is established by the results of our study where 10% GA substantially reduced the dentin surface microhardness.

In the present study, 10% GA was found to be more effective than CA in removing OrthoMTA from the root sections. This suggests that GA in higher concentrations can create more porosities in OrthoMTA and make it more susceptible to be dislodged using ultrasonics. On the contrary, 10% CA had less negative impact on the dentinal structure than GA. Findings from the present study are in the agreement with Eldeniz A.U. et al. who found citric acid to be less effective in reducing dentin microhardness when compared to EDTA [26]. Marshall Jr. G.W. et al. used atomic force microscopy to study the effect of citric acid on dentin demineralization, dehydration and rehydration processes [27]. Arslan H. et al. studied the effect of citric acid irrigation on the fracture resistance of endodontically treated roots and found it to be safe to use [28].

This study utilized 5 mm sections of roots primarily due to following reasons: firstly, to standardize the specimen length, and secondly, to mimic the clinical scenario as 4–5 mm of OrthoMTA is commonly used in the majority of clinical situations. However, 5 mm of a section can be carefully negotiated using ultrasonics; however, ultrasonics fail to remove the well-condensed and adapted MTA along the dentin walls, which is often penetrated within the intricacies of the root canal system. Hence, use of additional irrigating and dissolution agents accompanied by mechanical instruments is a vital approach that could be less destructive and may better preserve the residual dentin.

Further studies are needed to explore the effect of various concentrations of different acids, pH and exposure times of acidic solutions on OrthoMTA, microhardness and the strength of the radicular dentin surface. In addition, the effect of using ultrasonics accompanied by acidic solutions on the surface topography of dentin needs to be further examined using microscopic techniques. Furthermore, the biocompatibility of used acidic solutions needs to be addressed before the results can be translated to clinical scenarios.

## 5. Conclusions

Within the scope of this laboratory research, 10% GA combined with ultrasonics was the most effective method in removing OrthoMTA. However, both 5% and 10% GA substantially reduced the dentin microhardness. Further studies are needed to validate the results of this research.

## Figures and Tables

**Figure 1 materials-16-04535-f001:**
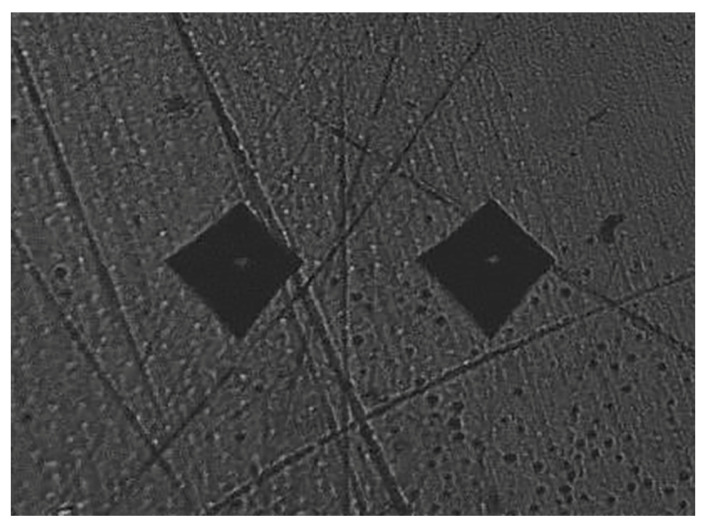
Representative surface microhardness testing image showing two diamond shaped indentations situated 1 mm or more apart over the tested dentin surface.

**Figure 2 materials-16-04535-f002:**
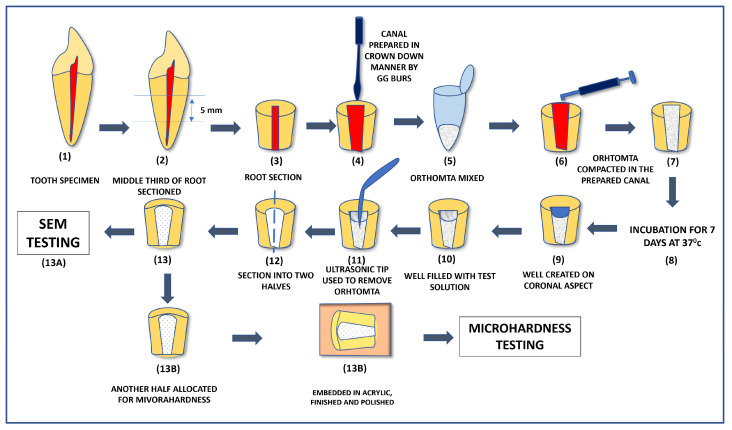
Schematic flow-chart diagram of the methodology showing the step-by-step procedure.

**Figure 3 materials-16-04535-f003:**
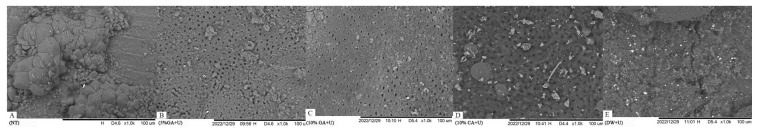
(**A**–**E**) Groupwise representative SEM images (1000×). (**A**) (NT) Entire dentin surface covered with OrthoMTA; (**B**) (5% GA+U) Substantially cleaned dentin surface visible in the form of open dentinal tubules along with fewer fragments of residual OrthoMTA visible; (**C**) (10% GA+U) Cleanest dentin surface visible with minimal presence of residual OrthoMTA in the form of smaller fragments; (**D**) (10% CA+U) Visible open dentin tubules suggestive of removal of OrthoMTA with presence of gross chunks of residual MTA; (**E**) (DW+U) Almost entire dentin surface covered with residual MTA with presence of cracks and irregularities over the residual OrthoMTA surface.

**Figure 4 materials-16-04535-f004:**
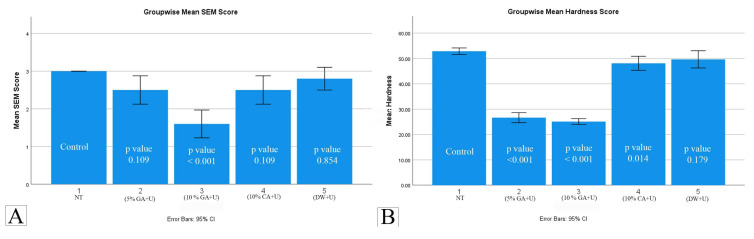
Histogram depicting groupwise mean values along with standard deviation, shown as error bars. (**A**) Residual OrthoMTA score. (**B**) Surface microhardness score.

**Table 1 materials-16-04535-t001:** Table showing the groupwise mean difference values of obtained SEM and surface microhardness tests.

Confidence Interval Set at 95%					
Dependent Variable	(I) Group	(J) Group	Mean Difference (I–J)	Std. Error	*p* Value
SEM	NT	5% GA+U	0.500	0.200	0.109
		10% GA+U	1.400 *	0.200	<0.001
		10% CA+U	0.500	0.200	0.109
		DW+U	0.200	0.200	0.854
	5% GA+U	NT	−0.500	0.200	0.109
		10% GA+U	0.900 *	0.200	<0.001
		10% CA+U	0.000	0.200	1.000
		DW+U	−0.300	0.200	0.568
	10% GA+U	NT	−1.400 *	0.200	<0.001
		5% GA+U	−0.900 *	0.200	<0.001
		10% CA+U	−0.900 *	0.200	<0.001
		DW+U	−1.200 *	0.200	<0.001
	10% CA+U	NT	−0.500	0.200	0.109
		5% GA+U	0.000	0.200	1.000
		10% GA+U	0.900 *	0.200	<0.001
		DW+U	−0.300	0.200	0.568
	DW+U	NT	−0.200	0.200	0.854
		5% GA+U	0.300	0.200	0.568
		10% GA+U	1.200 *	0.200	<0.001
		10% CA+U	0.300	0.200	0.568
Microhardness	NT	5% GA+U	26.20000 *	1.42441	<0.001
		10% GA+U	27.75000 *	1.42441	<0.001
		10% CA+U	4.77000 *	1.42441	0.014
		DW+U	3.21000	1.42441	0.179
	5% GA+U	NT	−26.20000 *	1.42441	<0.001
		10% GA+U	1.55000	1.42441	0.812
		10% CA+U	−21.43000 *	1.42441	<0.001
		DW+U	−22.99000 *	1.42441	<0.001
	10% GA+U	NT	−27.75000 *	1.42441	<0.001
		5% GA+U	−1.55000	1.42441	0.812
		10% CA+U	−22.98000 *	1.42441	<0.001
		DW+U	−24.54000 *	1.42441	<0.001
	10% CA+U	NT	−4.77000 *	1.42441	0.014
		5% GA+U	21.43000 *	1.42441	<0.001
		10% GA+U	22.98000 *	1.42441	<0.001
		DW+U	−1.56000	1.42441	0.808
	DW+U	NT	−3.21000	1.42441	0.179
		5% GA+U	22.99000 *	1.42441	<0.001
		10% GA+U	24.54000 *	1.42441	<0.001
		10% CA+U	1.56000	1.42441	0.808

* indicates statistically significant difference.

## Data Availability

Not applicable.

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
