# Peer review of "Effect of Various Acid Solutions as an Aid in Removing the OrthoMTA-Based Root Canal Filling"

_materials, 2023, doi:10.3390/ma16134535_

Round 1

Reviewer 1 Report

The research within manuscript entitled ,,Effect of various acid solutions as an aid in removing the OrthoMTA based root canal filling” is interesting and bring useful information to the specialists. Ultrasound treatment combined with various acid etching was employed to remove Ortho MTA remains from the root canal dentine. Experimental setup is well designed and implemented. Unfortunately the obtained results are presented in a poor manner and several unclear aspects occur. Therefore the manuscript must be revised according to the comments below:

Comment 1) Figure 1 contains some of the most important obtained results, therefore it must be moved from ,,Materials and methods” to ,,Results”.

Comment 2) SEM image for the control group (NT) is missing; it must be properly presented beside the other SEM images.

Comment 3) The microstructural aspects of each SEM image must be described and discussed in text.

Comment 4) Table 1 contains the statistical processing of the obtained values but it is not clear for the readers. Therefore, it must be removed and the obtained mean values must be presented as histogram plots:

- One histogram for the mean residual Ortho MTA coverage scores for each group;

- One histogram for the mean Vickers microhardness for each group.

The standard deviation must be displayed as error bars, and the relevant p values must be properly marked on the graphs.

Comment 5) Some of the references are wrong numbered in text, for example line 164 ,,Boutsioukis C et al.[10], Nandini S et al.[10] and Butt N et al.[11]” should be ,,Boutsioukis C et al.[8], Nandini S et al.[9] and Butt N et al.[10]”. All references in text must be revised to be in agreement with the reference list.

Moderate editing of English language is required. 

Author Response

Dear Reviewer,

Thank you for providing timely and vital suggestions. Please find the point by point amendments and explanation to all the comments as follows:

Point 1: Figure 1 contains some of the most important obtained results, therefore it must be moved from ,,Materials and methods” to ,,Results”.

Response 1: Figure 1A-1E. Groupwise representative SEM images (1000x) has been added and legends for the same are needfully described.

Point 2: SEM image for the control group (NT) is missing; it must be properly presented beside the other SEM images.

Response 2: (NT) representative image has been added to the Figure 1A.

Point 3:The microstructural aspects of each SEM image must be described and discussed in text.

Response 3: Microstructural aspects described in the first paragraph of the result section (first paragraph, line 156-165).

Point 4:Table 1 contains the statistical processing of the obtained values but it is not clear for the readers. Therefore, it must be removed and the obtained mean values must be presented as histogram plots:

- One histogram for the mean residual Ortho MTA coverage scores for each group;

- One histogram for the mean Vickers microhardness for each group.

The standard deviation must be displayed as error bars, and the relevant p values must be properly marked on the graphs.

Response 4: Additional histogram charts as suggested is added as figure 4 in the revised manuscript file. Please refer to Figure 4 at page 5. Groupwise mean difference values for SEM and surface microhardness scores are presented in Table 1.

Point 5: Some of the references are wrong numbered in text, for example line 164 ,,Boutsioukis C et al.[10], Nandini S et al.[10] and Butt N et al.[11]” should be ,,Boutsioukis C et al.[8], Nandini S et al.[9] and Butt N et al.[10]”. All references in text must be revised to be in agreement with the reference list.

Response 5: References are rechecked, corrected and cited accoridng to the journal’s guidleines as suggested.

Comments on the Quality of English Language: Moderate editing of English language is required.

Response 6: Entire manuscript is re-reviewed along with needful English language editing.

Reviewer 2 Report

Dear Authors,

1.     Your work is laboratory-experimental and the purpose is to translate the results into a clinical model. Therefore, it seems important to estimate the number of the study sample for statistical purposes. How did you estimate the sample size in your study?

2.     Line 88, the word "distiled" is unnecessarily capitalized

3.     Why is complete MTA removal so clinically important? If the tooth is referred for endodontic re-treatment and the MTA can be placed in layers, is it not enough to achieve proper root canal patency even if the remnants of the old filling still occlude some of the dentinal tubules?

4.     In your model you assumed that the samples are 5 mm. They are filled to 4 mm (as 1 mm is mechanically unblocked). Is the synergistic effect of the individual rinsing solutions the same regardless of the depth of preparation? Does the penetration of the solution decrease as you go deeper into the sample? If so, won't it be an obstacle when we clinically remove MTA from the entire length of the canal? However, if we assume that the thickness of MTA used clinically is usually lower, is it really a challenge to remove it?

5.     You state, that you used 60 samples divided into groups of 10 and you describe only 5 groups (lines 84-88). If the 60 tooth samples were divided into 5 groups of 10, what about the last 10 samples?

6.     No SEM images for group 1 ( Figure 2), need to be supplemented.

7.     Move the first sentence of the results to the Materials and Methods chapter (lines 122/123)

8.     In lines 123-127 you give the interpretation of the results, not the result. The results are no observations of MTA remnants. Your suggestions as to why this happened are the domain of the Discussion chapter.

9.     Line 132 illegible, no mention of group 3.

10.  Lines 134-7 contains an unauthorized statement due to the lack of statistical significance of differences between the groups - this mention should be deleted

11.  Table 1 illegible, please correct it.

12.  How is it that in table 1 you give many negative SEM scores if its scale is from 0 to 3 and therefore can never be negative?

13.  You described groups of samples and then you use either group names or compositions of lavage fluids interchangeably - you need to standardize these data in the text of the manuscript.

14.  Discussion: lines 152-162 repeat the introduction - please delete

15.  Conclusions are repetitions of results, not conclusions. Moreover, you provide results without statistical significance as conclusions, which may be misleading. It is necessary to remove statistically insignificant results from the conclusions.

Best regards

Dear Editor,

The manuscript has serious methodological flaws that need to be clarified and corrected before potential publication. I included detailed comments in a letter to the authors.

Best regards

Piotr Wychowański

Author Response

Dear Reviewer,

Thank you for providing timely and vital suggestions. Please find the point by point amendments and explanation to all the comments as follows:

Point 1: Your work is laboratory-experimental and the purpose is to translate the results into a clinical model. Therefore, it seems important to estimate the number of the study sample for statistical purposes. How did you estimate the sample size in your study?

 Response 1: Sample size estimation method has been described in the methodology section as requested. (Page 2, line 74-77)

Point 2: Line 88, the word "distiled" is unnecessarily capitalized

Response 2: Edited as suggested.

Point 3: Why is complete MTA removal so clinically important? If the tooth is referred for endodontic re-treatment and the MTA can be placed in layers, is it not enough to achieve proper root canal patency even if the remnants of the old filling still occlude some of the dentinal tubules?

Response 3: Explanantion has been given in the introduction section along with the supporting reference. (Page 2, line 48-51)

Point 4: In your model you assumed that the samples are 5 mm. They are filled to 4 mm (as 1 mm is mechanically unblocked). Is the synergistic effect of the individual rinsing solutions the same regardless of the depth of preparation? Does the penetration of the solution decrease as you go deeper into the sample? If so, won't it be an obstacle when we clinically remove MTA from the entire length of the canal? However, if we assume that the thickness of MTA used clinically is usually lower, is it really a challenge to remove it?

 Response 4: In the event of treating longer root canals inserting irrigation needle to the required depth and replenishing irrigating solution will be deemed necessary. Even in the lesser thickness of MTA, the role of dissolving agent is prudent as ultrasonics alone may be inadequate in removing the MTA specially when it is strongly adhering within the intricacies of the root canal walls. Explanantion has been given in the discussion section. (Page 8, Line 252-259)

Point 5: You state, that you used 60 samples divided into groups of 10 and you describe only 5 groups (lines 84-88). If the 60 tooth samples were divided into 5 groups of 10, what about the last 10 samples?

 Response 5: It was typographic error. Necessary amendments have been done to make it uniform sample saize 50.

Point 6: No SEM images for group 1 (Figure 2), need to be supplemented.

 Response 6: SEM image for group 1 has been added as Figure 1A.

Point 7: Move the first sentence of the results to the Materials and Methods chapter (lines 122/123)

 Response 7: Amended as suggested.

Point 8: In lines 123-127 you give the interpretation of the results, not the result. The results are no observations of MTA remnants. Your suggestions as to why this happened are the domain of the Discussion chapter.

 Response 8: The description of results has been removed from result section and moved to discussion.

Point 9: Line 132 illegible, no mention of group 3.

 Response 9: Necessary ammendments have been done.

Point 10: Lines 134-7 contains an unauthorized statement due to the lack of statistical significance of differences between the groups - this mention should be deleted

 Response 10: This statement has been deleted as suggested.

Point 11: Table 1 illegible, please correct it.

 Response 11: Table 1 has been modified. In addition, for better understanding of the readers the histogram with p values and standard error bar has been added in the result section.

Point 12: How is it that in table 1 you give many negative SEM scores if its scale is from 0 to 3 and therefore can never be negative?

 Response 12: Table 1 contains mean difference of obtained score rather than mean scores. Difference amongst the mean scores of two camparing groups is negative in certain situations. The table has been reformatted to include the significant values and legend for the table is also rewritten for the better understanding of the readers.

Point 13: You described groups of samples and then you use either group names or compositions of lavage fluids interchangeably - you need to standardize these data in the text of the manuscript.

 Response 13: Group description has been made unifrom across entire manuscript using lavage fluid initials.

Point 14: Discussion: lines 152-162 repeat the introduction - please delete

 Response 14: It has been deleted as suggested.

Point 15: Conclusions are repetitions of results, not conclusions. Moreover, you provide results without statistical significance as conclusions, which may be misleading. It is necessary to remove statistically insignificant results from the conclusions.

Response 15: Necessary amendments have been made in the conclusion section as suggested.

Reviewer 3 Report

Dear authors,

Hello,

Review:  Effect of various acid solutions as an aid in removing the OrthoMTA based root canal filling

 Abstract: Well written

Introduction: Well written, concise, well-chosen references

Material and methods:

- The working method was described in detail, the representative images

- Statistical analysis methods can be written in material and methods section instead of "Results"

Results:

- Statistical analysis methods can be passed to material and methods instead of "Results"

- Table 1 is not included in the text

Discussions

- Limitations of the study?

Author Response

Dear Reviewer,

Thank you for providing timely and vital suggestions. Please find the point by point amendments and explanation to all the comments as follows:

 Abstract: Well written

Response: Thank you. The word count for abstract is furhter reduced to limit as per journal’s guideline.

Introduction: Well written, concise, well-chosen references

Response: Thank you.

Material and methods:

- The working method was described in detail, the representative images

- Statistical analysis methods can be written in material and methods section instead of "Results"

Response: Necessary changes made as suggested.

Results:

- Statistical analysis methods can be passed to material and methods instead of "Results"

- Table 1 is not included in the text

Response: Necessary changes made as suggested.

Discussions

- Limitations of the study?

Response: Necessary changes made as suggested. Page 8, Line 260-266)

Reviewer 4 Report

Many thanks to the authors for the submission their. However some modifications are required in order to proceed to publication.

1- The abstract is longer than what the journal suggests, so please reduce the materials and methods. As such, I recommend you to kindly reframe the abstract. You also should identify the (MTA) in the abstract.

2- The introduction should be expand by including  some previous studies and also I recommend the authors to emphasis the hypothesis at the end of the introduction.

3- In line 71: size 2 to size 6, the authors should mention the unit.

4- In line 84: the authors claim that ' All specimens were randomly allocated to six groups ....' . However, they claim only 5 groups in the abstract. May I ask the authors where is Group 6?

5- In line 172.' Boutsioukis C et al.[10], Nandini S et al.[10]..'  how come those references are the same? Also, reference [11], doesn't match with the reference list. Please you should go through ALL the references from the Introduction to Discussion to make sure they are correct.

Thanks

Nothing serious.

Author Response

Dear Reviewer,

Thank you for providing timely and vital suggestions. Please find the point by point amendments and explanation to all the comments as follows:

Point 1: The abstract is longer than what the journal suggests, so please reduce the materials and methods. As such, I recommend you to kindly reframe the abstract. You also should identify the (MTA) in the abstract.

Response 1: The word count has been reduced substantially. Any further reduction in word count is difficult as it contains key information.

Point 2: The introduction should be expand by including  some previous studies and also I recommend the authors to emphasis the hypothesis at the end of the introduction.

Response 2: Additional information about the background of the study along with its references has been added to the introduction section. Hypothesis has been added at the end of introduction.

Point 3: In line 71: size 2 to size 6, the authors should mention the unit.

Response 3: It has been added as suggested. (Page 2 line 82-83)

Point 4: In line 84: the authors claim that ' All specimens were randomly allocated to six groups ....' . However, they claim only 5 groups in the abstract. May I ask the authors where is Group 6?

Response 4: It was a typographic error. Necessary amendments have been done to make it uniform sample saize 50.

Point 5: In line 172.' Boutsioukis C et al.[10], Nandini S et al.[10]..'  how come those references are the same? Also, reference [11], doesn't match with the reference list. Please you should go through ALL the references from the Introduction to Discussion to make sure they are correct.

Response 5: Corrections have been done as suggested. Reference numbering has been done accoriding to the journal’s guidleines.  

Round 2

Reviewer 1 Report

All requested major corrections and completions were well effectuated and the manuscript was properly improved.

Some minor issue still must to be fixed:

- Figure 1 must be moved from Materials and Methods to the Results - it is mandatory.

- English was considerably improved. Only some minor aspects must be revised, for example: the verb is missing in the sentence lines 7-9. 

Minor editing of English language required, for example: the verb is missing in the sentence lines 7-9.

Author Response

Dear reviewer,

Thank you for your kind recommendations. All the necessary changes are carried out in the manuscript file keeping the track changes option on in MS word. Additionally explained as follows:

Comment 1: 

Figure 1 must be moved from Materials and Methods to the Results - it is mandatory.

Answer: Changes made as suggested. The figure 1 is now figure 3 and moved to results section.

Comment 2:

The verb is missing in the sentence lines 7-9. 

The line 7-9 are reframed as pert he requirements.

Looking forward to hearing from you soon.  

Regards,

Reviewer 2 Report

Dear Authors,

I am fully satisfied with Your replies to my comments and manuscript modifications made.

In my opinion it is ready for publication in the present form.

Best regards

Author Response

Dear Reviewer,

Thank you for your valuable suggestions and recommendations.

Looking forward,

Regards,

Reviewer 4 Report

Thank you for the modification that you have made. However, there is only two points need to be considered before final acceptance:

1- The authors did not identify the (MTA) in the abstract as I recommended.

2- line 82-83, is there any unit instead of ISO size? what does ISO size mean? 

Thanks

Minor editing needed.

Author Response

Dear reviewer,

Thank you for your kind recommendations. All the necessary changes are carried out in the manuscript file keeping the track changes option on in MS word. Additionally explained as follows:

Comment 1: 

The authors did not identify the (MTA) in the abstract as I recommended.

Answer: Changes made as suggested. The MTA and its full form mentioned in the abstract.

Comment 2:

Line 82-83, is there any unit instead of ISO size? what does ISO size mean? 

Answer: 

Necessary description in the measurement values as millimeter mentioned in the section.

Thank you.

Looking forward to hearing form you soon.  

Regards,
